# Hippocampal subfield volumes and pre-clinical Alzheimer's disease in 408 cognitively normal adults born in 1946

**Thomas D. Parker**[1], **David M. Cash**[1], **Christopher A. S. Lane**[1], **Kirsty Lu**[1], **Ian B. Malone**[1], **Jennifer M. Nicholas**[1,2], **Sarah-Naomi James**[3], **Ashvini Keshavan**[1], **Heidi Murray-Smith**[1], **Andrew Wong**[3], **Sarah M. Buchanan**[1], **Sarah E. Keuss**[1], **Carole H. Sudre**[1,4,5], **Marc Modat**[1,4,5], **David L. Thomas**[6,7], **Sebastian J. Crutch**[1], **Marcus Richards**[3], **Nick C. Fox**[1], **Jonathan M. Schott**[1]*

1 The Dementia Research Centre, Queen Square Institute of Neurology, University College London, London, United Kingdom, 2 Department of Medical Statistics, London School of Hygiene and Tropical Medicine, London, United Kingdom, 3 MRC Unit for Lifelong Health and Ageing at University College London, London, United Kingdom, 4 School of Biomedical Engineering and Imaging Sciences, King's College London, London, United Kingdom, 5 Department of Medical Physics and Biomedical Engineering, University College London, London, United Kingdom, 6 Leonard Wolfson Experimental Neurology Centre, Queen Square Institute of Neurology, University College London, London, United Kingdom, 7 Neuroradiological Academic Unit, Department of Brain Repair and Rehabilitation, Queen Square Institute of Neurology, University College London, London, United Kingdom

* j.schott@ucl.ac.uk

## Abstract

### Background

The human hippocampus comprises a number of interconnected histologically and functionally distinct subfields, which may be differentially influenced by cerebral pathology. Automated techniques are now available that estimate hippocampal subfield volumes using *in vivo* structural MRI data. To date, research investigating the influence of cerebral β-amyloid deposition—one of the earliest hypothesised changes in the pathophysiological continuum of Alzheimer's disease—on hippocampal subfield volumes in cognitively normal older individuals, has been limited.

### Methods

Using cross-sectional data from 408 cognitively normal individuals born in mainland Britain (age range at time of assessment = 69.2–71.9 years) who underwent cognitive assessment, [18]F-Florbetapir PET and structural MRI on the same 3 Tesla PET/MR unit (spatial resolution 1.1 x 1.1 x 1.1. mm), we investigated the influences of β-amyloid status, age at scan, and global white matter hyperintensity volume on: CA1, CA2/3, CA4, dentate gyrus, presubiculum and subiculum volumes, adjusting for sex and total intracranial volume.

### Results

Compared to β-amyloid negative participants (n = 334), β-amyloid positive participants (n = 74) had lower volume of the presubiculum (3.4% smaller, p = 0.012). Despite an age range at scanning of just 2.7 years, older age at time of scanning was associated with lower CA1

**Data Availability Statement:** Data cannot be shared publicly in line with data sharing policies specific to the MRC National Survey of Health and Development to protect confidentiality of the study

participants. Data are available from https://www.nshd.mrc.ac.uk/data/ for researchers who meet the criteria for access to confidential data.

**Funding:** This study is principally funded by grants from Alzheimer's Research UK (ARUK-PG2014-1946, ARUK-PG2017-1946), the Medical Research Council Dementias Platform UK (CSUB19166), and the Wolfson Foundation (PR/ylr/18575). The genetic analyses are funded by the Brain Research Trust (UCC14191). Florbetapir amyloid tracer is kindly provided by Avid Radiopharmaceuticals (a wholly owned subsidiary of Eli Lilly) who had no part in the design of the study: we are particularly indebted to the support of the late Dr Chris Clark of Avid Radiopharmaceuticals who championed this study from its outset. The NSHD, MR, and AW are funded by the Medical Research Council (MC_UU_12019/1, MC_UU_12019/2, MC_UU_12019/3). Some researchers are supported by the NIHR Queen Square Dementia BRU (JMS, NCF), UCL Hospitals Biomedical Research Centre (JMS), Leonard Wolfson Experimental Neurology Centre (JMS, NCF). TDP is supported by a Wellcome Trust Clinical Research Fellowship (200109/Z/15/Z). CHS is supported by an Alzheimer's Society Junior Fellowship (AS-JF-17-011). SJC is supported by an Alzheimer's Research UK Senior Research Fellowship. NCF acknowledges support from the MRC, the UK Dementia Research Institute at UCL, and an NIHR Senior Investigator award, and additional funding from the EPSRC. JMS acknowledges the EPSRC (EP/J020990/1), BHF (PG/17/90/33415), Weston Brain Institute (UB170045), and European Union's Horizon 2020 research and innovation programme (Grant 666992).

**Competing interests:** NCF's research group has received payment for consultancy or for conducting studies from Biogen, Eli Lilly Research Laboratories, GE Healthcare, and Roche. NCF receives no personal compensation for the activities mentioned above. JMS has received research funding from Avid Radiopharmaceuticals (a wholly owned subsidiary of Eli Lilly), has consulted for Roche Pharmaceuticals, Biogen, Merck and Eli Lilly, given educational lectures sponsored by GE Healthcare, Eli Lilly and Biogen, and serves on a Data Safety Monitoring Committee for Axon Neuroscience SE. This does not alter our adherence to PLOS ONE policies on sharing data and materials.

**Abbreviations:** BaMoS, Bayesian Model Selection; CA, Cornu ammonis; HATA, Hippocampal amygdala transition area; MCI, Mild cognitive impairment; MMSE, Mini mental state examination; MRI, Magnetic Resonance Imaging; NSHD, MRC

(p = 0.007), CA4 (p = 0.004), dentate gyrus (p = 0.002), and subiculum (p = 0.035) volumes. There was no evidence that white matter hyperintensity volume was associated with any subfield volumes.

## Conclusion

These data provide evidence of differential associations in cognitively normal older adults between hippocampal subfield volumes and β-amyloid deposition and, increasing age at time of scan. The relatively selective effect of lower presubiculum volume in the β-amyloid positive group potentially suggest that the presubiculum may be an area of early and relatively specific volume loss in the pathophysiological continuum of Alzheimer's disease. Future work using higher resolution imaging will be key to exploring these findings further.

## Introduction

Hippocampal atrophy is a characteristic feature of Alzheimer's disease (AD) and also occurs to a lesser extent in ageing [1–8]. The hippocampus comprises interconnected histologically and functionally distinct subfields and delineation of these subfields from structural magnetic resonance imaging (MRI) has the capacity to provide new insights into mechanisms of disease [9]. There is evidence suggesting hippocampal subfields may be differentially influenced by AD, vascular disease, and ageing [10–25] and have the potential to be more specific biomarkers of neurodegeneration in ageing populations. However, to date, research investigating the relationships between cerebral β-amyloid deposition and the volume of individual hippocampal subfields in cognitively normal older individuals has been limited. In a small sample (n = 74), Hsu and colleagues reported β-amyloid associated decreases in not only total hippocampal volume, but also the subiculum and pre-subiculum [26]. The segmentation algorithm utilized in this study has been shown to be vulnerable to mislabelling [27] and there is a requirement for studies with larger sample sizes and more up to date hippocampal subfield segmentation methodology to further investigate this relationship.

We report a cross-sectional analysis of a large sample of individuals all born in mainland Britain in the same week of March 1946 who underwent cognitive assessment, [18]F-Florbetapir positron emission tomography (PET) and structural MRI aged 69.2–71.9 years. Utilizing an updated version of Freesurfer's hippocampal subfield segmentation tool based on a computational atlas using *ex vivo*, ultra-high resolution MRI, the objective of this analysis was to investigate the hypothesis that individual hippocampal subfield volumes are differentially associated with β-amyloid deposition, age at time of scan and global white matter hyperintensity volume (WMHV–a surrogate marker of cerebral small vessel disease).

## Materials and methods

### Participants

Data were analysed from individuals who participated in Insight-46, a neuroscience sub-study of the MRC National Survey of Health and Development (NSHD). The NSHD originally comprised 5362 individuals all born in mainland Britain in one week of March 1946 [28–30]. Insight-46 recruited 502 participants to a single-site study involving detailed clinical and neuropsychological assessment, MRI and [18]F-florbetapir PET imaging [31,32] conducted over a 2.7 year period. Ethical approval was granted by the National Research Ethics Service

National Survey of Health and Development; PET, Positron Emission Tomography; SUVR, Standard uptake value ratio; TIV, Total intracranial volume; WMHV, White matter hyperintensity volume.

Committee London (reference 14/LO/1173). All participants provided written informed consent. Exclusions from the analysis were: failure to complete scan (n = 31); PET acquisition failure (n = 8); failure of WMHV segmentation (n = 4); movement artefact felt to impact reliability of hippocampal subfield segmentation (n = 3); and participants with evidence of dementia, MCI, major neurological or psychiatric disorder (n = 48).

## Clinical assessment

As part of Insight-46 Each participant underwent a history of cognitive impairment, major neurological or psychiatric illness. Participant cognitive concern was defined as self-report of memory or cognitive difficulties more than others the same age, or if they felt they would seek medical attention regarding cognitive difficulties. An informant history regarding each participant's cognitive functioning was acquired using the AD8 questionnaire [33,34]. Informant cognitive concern was defined as an AD8 score ≥ 2.

Cognitive testing included: the Mini-Mental State Examination (MMSE) [35]; the digit-symbol substitution test [36]; logical memory delayed recall [37]; matrix reasoning [38]; and the 12-item Face-Name test [31,39].

Participants were defined as having mild cognitive impairment (MCI) if there was evidence of significant cognitive concerns from the participant or the informant AND objective evidence of an amnestic (sample specific logical memory delayed recall score cut-off ≥ 1.5 standard deviations below the mean) or non-amnestic cognitive deficit (sample specific digit-symbol substitution score cut-off ≥ 1.5 standard deviations below the mean), AND there was no evidence of dementia. Logical memory delayed recall and digit-symbol substitution were selected for this as they both exhibited a normal distribution.

## Florbetapir-PET

Concurrent acquisition of PET and MRI was performed on the same Siemens Biograph mMR 3 Tesla PET/MRI scanner. Static PET images representing uptake during a 10-minute period approximately 50 minutes after injection of approximately 370 megabecquerels of [18]F-florbetapir were reconstructed using a thorughly validated pseudo-CT attenuation correction method [40]. The post-uptake images were then rigidly registered to the structural MRI data using a symmetric block matching technique [41]. A previously defined cortical grey matter composite (composed of frontal, temporal, parietal, and cingulate regions [42–46]) was selected as the primary region of interest to assess β-amyloid burden. All voxels in the image were then normalised to a reference region to produce a Standard Uptake Value Ratio (SUVR) image. The reference region selected was a mask of subcortical white matter, eroded one time to avoid partial volume effects. The advantages of the subcortical white matter as a reference region are that it is a relatively uniform tissue type and has less risk of corruption from other tissues compared to other commonly used reference regions (e.g. the cerebellum). Furthermore, SUVR values derived from a white matter reference region have also been shown to correlate better with gold standard arterial sampling based dynamic measurements of tracer uptake compared to SUVR values derived using other widely used reference regions (e.g. the cerebellum)[47]. Gaussian mixture models were used to fit the data and obtain a threshold for β-amyloid positivity. Mixture models with one, two, and three gaussians, were tested, with the best model selected using Bayesian Information Criteria. The best fit for the composite cortical grey matter region of interest was two gaussians. The 99[th] percentile SUVR value of the Gaussian representing the β-amyloid negative population was selected as the cut-point (0.6104) for β-amyloid positivity,

## Structural MRI

MRI sequences included: three-dimensional T1-weighted MPRAGE images (voxel size 1.1x1.1x1.1 mm$^3$ isotropic; TE/TR = 2.92/2000, total time = 5 minutes 6 seconds) and three-dimensional FLAIR images using an IR-SPACE acquisition scheme (voxel size 1.1 x1.1x1.1 mm$^3$ isotropic; TE/TR = 402/5000, total time = 6 minutes 27 seconds) [31].

All MRI data were preprocessed for gradwarp and image inhomogeneity [48,49]. Furthermore, all MRI data underwent a detailed quality control process by trained assessor to ensure there was adequate coverage and absence of motion artefact. T1 scans were also assessed for blurring, image wrap-around and contrast problems, and FLAIR for adequate CSF suppression [31].

Hippocampal subfield segmentation was performed using Freesurfer version 6.0; an algorithm based on a computational atlas using *ex vivo*, ultra-high resolution MRI that segments T1-weighted MRI data to the following subfields: CA1, CA2/3, CA4, fimbria, the hippocampal fissure, presubiculum, subiculum, hippocampal tail, parasubiculum, the molecular and granule cell layers of the dentate gyrus (referred to as the "dentate gyrus' for the remainder of the article), the molecular layer and the hippocampal amygdala transition area (HATA) [9]. Visual inspection of each participant's hippocampal subfield segmentation and corresponding T1-weighted structural MRI data was performed to ensure each segmentation conformed to the hippocampus and there were no clear and obvious errors. This was performed with the caveat that the that precise visualisation of the boundaries that define the distinct hippocampal subfields at the spatial resolution used in this study is challenging [50]. The following regions were excluded prior to the analysis: the hippocampal fissure (a thin CSF layer rather than a hippocampal substructure *per se)*, the molecular layer (a thin white matter layer, which is at risk of partial volume effects), the fimbria (small volume white matter region, also at risk of partial volume effect), the parasubiculum and HATA (both of which have volumes <100 μl and may be more prone to noise), and the hippocampal tail (which is not a histologically distinct region, but instead represents a conglomeration of CA1-4 and dentate gyrus) [9]. The hippocampal subfield segmentation and corresponding T1-weighted structural images for each participant were visually inspected to exclude major errors. For each subfield investigated a total was calculated by summing the left and right hemisphere volumes.

Bayesian Model Selection (BaMoS) [51], an automated segmentation tool that uses T1-weighted and FLAIR MRI data, was used to generate a global estimate of WMHV.

Total intracranial volume (TIV) was calculated from the T1-weighted images using statistical parametric mapping 12 software [52].

## Statistical approach

Two-sample t-tests, or where there was a material departure from a normal distribution, Wilcoxon rank sum tests, were used to compare continuous clinical and cognitive characteristics between β-amyloid positive and negative groups. Logistic regression models were used to compare categorical variables between β-amyloid positive and negative groups. Spearman's correlation coefficients were used to assess unadjusted relationships between age at time of scanning and continuous variables. Wilcoxon rank sum tests were used to test associations between age at scanning and categorical variables.

Linear regression models with robust standard errors were used to test the hypothesis that individual hippocampal subfield volumes (dependent variables) are associated with: β-amyloid status, age at time of scan and global WMHV (predictor variables of interest), with additional adjustment for sex and TIV [52]. Following linear regression analysis, mean differences in subfield volumes between β-amyloid positive and negative participants were expressed as a

percentage of the mean total volume of the respective subfield in the β-amyloid negative population. The β-coefficient for age at scan (i.e. estimated difference in volume per year) was expressed as a percentage of the mean total volume of the respective subfield across the whole sample.

A threshold for statistical significance of p<0.05 was utilized throughout the analysis.

## Results

### Sample characterisation

As individuals were born in the same week, differences in age within the sample were due to the date of assessment and were therefore narrow: median = 70.7 years; range = 69.2–71.9 years. There were no statistically significant differences between β-amyloid positive and negative individuals in the age, WMHV, sex, TIV, logical memory, digit-symbol substitution scores and the 12-item Face-Name test. Matrix reasoning scores were lower in the β-amyloid positive group compared to the β-amyloid negative group (p = 0.037). There was a non-significant trend for lower MMSE scores in the β-amyloid positive group compared to the β-amyloid negative group (p = 0.063). As would be expected [53], *APOE* genotype strongly predicted Aβ-status, with $\epsilon$4 carriers being 5.22 times more likely to be β-amyloid positive. Age at time of scan was not associated with sex, TIV, or performance on any neuropsychological tests. There was evidence of a positive association between older age at time of scan and WMHV (p = 0.013) (Table 1).

### Influence of β-amyloid deposition

Compared to β-amyloid negative participants (n = 334), β-amyloid positive participants (n = 74) had lower presubiculum volume (3.4% smaller relative to the mean volume in the β-amyloid negative population, p = 0.012) independent of age at time of scan, global WMHV, sex and TIV (Table 2). β-amyloid-associated differences in CA1 (1.3%), CA2/3 (0.4%), CA4 (0.1%), dentate gyrus (0.3%) and subiculum (1.8%) volumes were directionally consistent but not significant (Table 2), while there was evidence of lower total hippocampal volume in the β-amyloid positive participants at trend level of significance only (1.6% smaller p = 0.075) (Table 2).

To explore asymmetry, post-hoc analysis examining the left and right hemispheres separately in the presubiculum was performed and revealed similar β-amyloid-associated differences in both the left (3.6% decrease, p = 0.023) and right presubiculum (3.3% decrease, p = 0.02). To explore whether the association observed between β-amyloid and presubiculum volume was driven by cognitively impaired individuals not captured by the MCI diagnostic criteria used to characterise the sample, a post-hoc regression analysis between presubiculum volume and the 12-item Face-Name test (i.e. a test of episodic memory not used in the diagnostic formulation in this study) was performed. There was no evidence that 12-item Face-Name test performance predicted presubiculum volume (p = 0.82), nor was there evidence of an interaction between 12-item Face-Name test performance and β-amyloid status in terms of its effect on presubiculum volume (p = 0.30).

### Influence of age at time of scanning

Older age at time of scan was associated with smaller volumes of CA1 (1.9%/year, p = 0.007), CA4 (1.7%/year, p = 0.004), dentate gyrus (2.0%/year, p = 0.002), subiculum (1.6%/year, p = 0.035) and total hippocampus (1.5%/year, p = 0.012) independent of β-amyloid status, global WMHV, sex and TIV. There was a non-significant trend for a negative association in

**Table 1. Sample characterisation–unadjusted relationships between clinical, demographic and cognitive outcomes with β-amyloid positivity and age at time of scan.**

| | β-amyloid negative (n = 334) | β-amyloid positive (n = 74) | β-amyloid negative vs positive | Association with age (n = 408) |
|---|---|---|---|---|
| Age, years, median (IQR) | 70.7 (1.2) | 70.7 (1.1) | p = 0.66[a] | - |
| Male sex, n (%) | 166 (49.7%) | 40 (54.1%) | OR 0.84; p = 0.5[c] | Δ = -0.05; p = 0.48[a] |
| MMSE, median (IQR) Maximum score = 30 | 30 (1) | 29 (1) | p = 0.063[a] | ρ = 0.0065; p = 0.9[b] |
| Logical memory score, mean (SD) Maximum score = 25 | 11.7 (3.6) | 11.3 (3.7) | p = 0.33[d] | ρ = 0.018; p = 0.72[b] |
| Digit-symbol substitution score, mean (SD) Maximum score = 93 | 48.8 (10.1) | 46.9 (9.7) | p = 0.14[d] | ρ = -0.014; p = 0.78[b] |
| Matrix reasoning, median (IQR) Maximum score = 32 | 26 (4) | 25 (4) | p = 0.037[a] | ρ = -0.073; p = 0.14[b] |
| 12-item Face-Name test, median (IQR) Maximum score = 96 | 66 (28) | 68 (27) | p = 0.29[a] | ρ = -0.063; p = 0.21[b] |
| *APOE ε4* carrier, n (%) (missing data: n = 2) | 76 (22.9%) | 45 (60.8%) | OR 5.22; p<0.0001[c] | p = 0.49[a] |
| TIV, mls, mean (SD) | 1426 (133) | 1451 (128) | p = 0.14[d] | ρ = 0.025; p = 0.61[b] |
| WMHV, mls, median (IQR) | 2.85 (4.84) | 3.30 (4.97) | p = 0.48[a] | ρ = 0.12; p = 0.013[b] |

[a]Wilcoxon rank sum test

[b]Spearman's rank correlation

[c]logistic regression

[d]t-test

Δ = mean difference; IQR = interquartile range; MMSE = mini-mental state examination; OR = odds ratio; ρ = Spearman's rho; SD = standard deviation; SUVR = standard uptake value ratio; TIV = total intracranial volume; WMHV = white matter hyperintensity volume (mls).

CA2/3 (1.4%/year, p = 0.086); while presubiculum volume was not significantly associated with age at time of scan (Fig 1 and Table 3).

To explore whether any associations specifically observed between age at time of scan and hippocampal subfield volumes might be explained by bias in recruitment order, analyses incorporating a wider range of co-variates that had been shown to predict recruitment in previous Insight-46 analyses (socioeconomic position, educational attainment, and childhood cognitive ability [32]) were also performed and made no material difference to the statistically significant results obtained (Table 4).

Socioeconomic position dichotomized into manual or non-manual based on occupation at 53 years (or earlier if missing) [56]

## Influence of WMHV

There was no evidence that WMHV predicted any hippocampal subfield volumes (Table 3).

## Discussion

Few studies have explored the extent to which cerebral β-amyloid deposition are associated with hippocampal subfield volume in cognitively normal older adults. In a large sample of

**Table 2. Independent influence of amyloid positivity on individual hippocampal subfield volumes in cognitively normal Insight-46 participants (n = 408) using linear regression models with robust standard errors (co- variates = age at scan, WMHV, sex and TIV).**

| | β-amyloid negative (n = 334) | β-amyloid positive (n = 74) | Absolute mean difference between β-amyloid negative and positive (95% CI) | %mean difference between β-amyloid negative and positive | p |
|---|---|---|---|---|---|
| CA1 | 1196 (141) | 1195 (111) | -15.4 (-38.6, 7.9) | 1.3% | 0.19 |
| CA2/3 | 406 (53) | 410 (47) | -1.5 (-12.3, 9.3) | 0.4% | 0.78 |
| CA4 | 482 (50) | 487 (44) | -0.6 (-10.4, 9.3) | 0.1% | 0.91 |
| Dentate gyrus | 554 (60) | 558 (53) | -1.8 (-13.5, 10.0) | 0.3% | 0.77 |
| Presubiculum | 582 (68) | 568 (67) | -19.9 (-35.4, -4.5) | 3.4% | **0.012**\* |
| Subiculum | 832 (100) | 827 (86) | -15.0 (-34.4, 4.5) | 1.8% | 0.13 |
| Total volume | 6515 (659) | 6487 (526) | -101.3 (-213.0, 10.3) | 1.6% | 0.075 |

CA = Cornu ammonis; TIV = total intracranial volume; WHMV = white matter hyperintensity volume

\*p<0.05

cognitively normal older individuals aged 69.2–71.9 years, we show differential associations between hippocampal subfield volumes β-amyloid deposition, and age at time of scan.

β-amyloid positive individuals had lower presubiculum volumes compared to β-amyloid negative individuals. Importantly, this was independent of age, sex, TIV and WMHV (a surrogate marker of cerebral small vessel disease [51]).

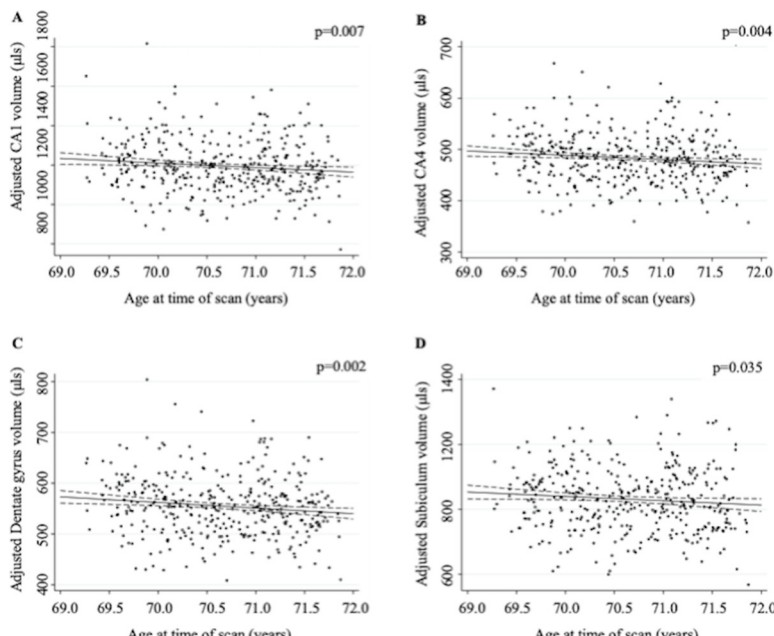

**Fig 1.** Age is associated with lower CA1 (panel A), CA4 (panel B), dentate gyrus (panel C) and subiculum (panel D) volume in cognitively normal older adults following adjustment for sex, TIV, amyloid status and WMHV. Dashed lines represent 95% confidence intervals, TIV = total intracranial volume; WMHV = white matter hyperintensity volume.

**Table 3. Independent influence of age at time of scan and WMHV on individual hippocampal subfield volumes in cognitively normal Insight-46 participants (n = 408) using linear regression models with robust standard errors (co- variates = β-amyloid status, sex and TIV).** Unstandardised β-coefficient for age represents mean subfield volume difference in μl per year. Unstandardised β-coefficient for WMHV represents mean subfield volume difference in μl per ml of WMHV.

| | Total volume, μl (SD) | Increasing age at time of scan (μl/year) | | % difference in volume per one year of age | WMHV (μl/ml) | |
|---|---|---|---|---|---|---|
| | | β-coefficient (95% CI) | p | | β-coefficient (95% CI) | p |
| CA1 | 1196 (136) | -23.0 (-39.8, -6.3) | **0.007***  | 1.9% | -0.59 (-2.35, 1.17) | 0.51 |
| CA2/3 | 407 (52) | -5.7 (-12.1, 0.8) | 0.086 | 1.4% | 0.06 (-0.62, 0.74) | 0.86 |
| CA4 | 483 (49) | -8.4 (-14.0, -2.7) | **0.004***  | 1.7% | -0.13 (-0.8, 0.54) | 0.7 |
| Dentate gyrus | 555 (59) | -11.1 (-17.9, -4.2) | **0.002***  | 2.0% | -0.35 (-1.17, 0.47) | 0.4 |
| Presubiculum | 580 (68) | -5.9 (-14.6, 2.8) | 0.18 | 1.0% | -0.2 (-1.24, 0.84) | 0.71 |
| Subiculum | 831 (98) | -13.5 (-25.9, -1.0) | **0.035***  | 1.6% | -0.7 (-2.17, 0.77) | 0.35 |
| Tail | 1047 (133) | -4.9 (-23.9, 14.1) | 0.61 | 1.5% | -0.07 (-2.21, 2.21) | 0.95 |
| Total volume | 6510 (637) | -99.6 (-176.9, -22.3) | **0.012***  | 1.9% | -3.13 (-11.67, 5.42) | 0.47 |

CA = Cornu ammonis; TIV = total intracranial volume; WHMV = white matter hyperintensity volume

*p<0.05

**Table 4. Additional adjustment for wide range of potential confounders that could be related to recruitment order makes no material difference to associations observed between hippocampal subfield volumes and age at time of scan.**

| | Increasing age (μl/year) Co-variates = sex, TIV, β-amyloid status, and WMHV | | Increasing age (μl/year) Co-variates = sex, TIV, β-amyloid status, WMHV, socioeconomic position, education, childhood cognition | |
|---|---|---|---|---|
| | β-coefficient (95% CI) | p | β-coefficient (95% CI) | p |
| CA1 | -23.0 (-39.8, -6.3) | **0.007***  | -23.1 (-40.0, -6.2) | **0.008***  |
| CA2/3 | -5.7 (-12.1, 0.8) | 0.086 | -6.0 (-12.5, 0.6) | 0.074 |
| CA4 | -8.4 (-14.0, -2.7) | **0.004***  | -8.7 (-14.4, -3.0) | **0.003***  |
| Dentate gyrus | -11.1 (-17.9, -4.2) | **0.002***  | -11.2 (-18.2, -4.4) | **0.001***  |
| Presubiculum | -5.9 (-14.6, 2.8) | 0.18 | -6.4 (-15.1, 2.2) | 0.15 |
| Subiculum | -13.5 (-25.9, -1.0) | **0.035***  | -13.8 (-26.3, -1.4) | **0.03***  |
| Total volume | -99.6 (-176.9, -22.3) | **0.012***  | -99.1 (-176.1, -22.2) | **0.012***  |

CA = Cornu ammonis; TIV = total intracranial volume; WHMV = white matter hyperintensity volume

***p<0.05.** Childhood cognitive function was measured at age 8 (or age 11 or 15 if this was missing) as the sum of scores of four tests of verbal and non-verbal ability standardised into a z-score [54]. Educational attainment was dichotomized into those with advanced (e.g. 'A level') or higher (e.g. university) qualifications, versus those below this level [55].

Research into hippocampal subfield volumes in pre-clinical populations previously has been limited. Hsu and colleagues reported β-amyloid associated decreases in the subiculum and pre-subiculum in a small sample of cognitively normal individuals (n = 74) [26], the pattern of which is broadly consistent with the findings presented in this analysis. However, the magnitude of these volume differences was much greater (approximately 10–12% compared to 3–4% in Insight 46). A number of factors including smaller sample size and wide age range (with a trend for the β-amyloid positive to be older) as well recruitment via convenience sampling in the previous study may account for these discrepancies. Furthermore, Hsu and colleagues utilized a previous version of Freesurfer's hippocampal subfield segmentation algorithm, which has been shown to be vulnerable to mislabelling [27].

Although studies across the pathophysiological continuum of AD suggest that atrophy of the presubiculum may be one of the earliest hippocampal anatomical markers of AD [15,20], no other studies to date, have demonstrated lower presubiculum volumes in relative isolation in β-amyloid positive cognitively normal individuals. Although the effect size was small, it was consistent across cerebral hemispheres suggesting both that this is a symmetrical effect, and also is unlikely to be the result of a type I error.

Although there was a trend for lower total hippocampal volume in our study, this did not attain a 5% level of significance. Previous cross-sectional studies focusing on total hippocampal volume alone have reported evidence of reduced baseline hippocampal volume associated with cerebral β-amyloid deposition in cognitively normal older individuals [20,57–62], whereas others have not [63,64]. Many studies reporting detectable differences in hippocampal volume include participants who are much older than the age range tested in this study and consequently are likely to be further along the pathophysiological continuum of AD and to have undergone greater levels of neuronal loss. Our data potentially suggest that individual hippocampal subfields may decrease in volume at an earlier stage of the pathophysiological continuum of AD before significant total hippocampal volume loss is detectable.

We also found that several different hippocampal regions–namely CA1, CA4, dentate gyrus and the subiculum–had a negative association with increasing age at time of scan. Previous work using a range of techniques has identified similar findings with age-associated volume loss in CA1 and the dentate gyrus [65]; CA1, dentate gyrus and CA4 [18]; subiculum and dentate gyrus [66]; CA1 and CA2 [67]. In addition to biological ageing effects, it is also possible that the age-related observations in this study might reflect differences in the characteristics of participants based on the order of recruitment. However, supplementary analyses incorporating a wide range of confounders that may influence study recruitment [32] made no difference to the associations observed between hippocampal subfield volumes and age at time of scan. It is unlikely technical factors (e.g. scanner drift), would explain the results given the lack of a relationship between age at time of scan and TIV (a morphometric measurement that would be expected to stay relatively stable across the age range studied). Additionally, the magnitude of the association between hippocampal subfields and age at time of scan were not dissimilar to estimated rates of change in hippocampal volume derived from studies where healthy older adults have been scanned multiple times over a short time interval [8].

Notably, the presubiculum was not associated with age at time of scan, even at a trend level, suggesting the β-amyloid associated effect of lower presubiculum volume observed in the β-amyloid positive group is unlikely to be related to ageing effects. Furthermore, neuropathological data suggests that the presubiculum is a site of large, evenly distributed "lake-like" β-amyloid deposits [68,69] in AD, but devoid of neurofibrillary tau deposition [69,70]. This might explain both the relatively selective β-amyloid associated volume difference in the presubiculum, as well as the relative sparing of an association with age at time of scan which has been

hypothesised to underpinned by primary age-related tauopathy a process thought to be independent of β-amyloid deposition [71,72].

Previous studies have shown that WMHV and cerebrovascular disease are important determinants of hippocampal atrophy in older adults [22,73]. However, we found no evidence that WMHV was associated with hippocampal subfield volumes. This may suggest that, at least in cognitively normal individuals aged approximately 70 years old, vascular disease burden as estimated by WMHV does not significantly influence cross-sectional hippocampal volume.

One significant limitation of this study is the spatial resolution (1.1 mm x 1.1 m x 1.1 m) provided by the volumetric structural T1-weighted MRI acquisition protocol utilized, as at this resolution boundaries important for subfield demarcation are not entirely visible and the computational atlas based on high resolution *ex vivo* data, relies on prior encoded information to provide volumetric estimates of individual hippocampal subfields [9]. It is clear that higher resolution imaging studies [14,18,74,75] are an important are of research. In particular, large scale comparison of results from hippocampal subfield segmentation protocols derived from widely used image acquisition protocols, such as the ones provided by this study, with higher resolution, but less widely implemented, imaging protocols are required. Another limitation of the study is the lack of a biomarker of neurofibrillary tangle deposition. A recent study of 88 individuals with a family history of AD investigating hippocampal sub-region volumes found those with abnormal CSF β-amyloid, but normal CSF tau had increased right subiculum volumes, whilst abnormal CSF β-amyloid and abnormal CSF tau was associated with decreased right subiculum volume [21] suggesting the presence of tau deposition is important for hippocampal atrophy and has selective effects on certain subfields. Future studies of hippocampal subfields employing tau PET imaging [76,77] will be of interest, particularly as tau is likely to influence hippocampal structure in both an β-amyloid dependent [77] and independent manner [71,72]. In addition, formal correction for multiple comparisons was not performed in this analysis, although efforts were made to constrain the analysis to total subfield volumes (sum of left and right) in the first instance and excluded subfields that were likely to be unreliable. Furthermore, to what extent hippocampal subfields are truly independent, and therefore to what extent it is valid to apply such techniques is unclear [16]. Finally, the cross-sectional nature of this study is a further limitation and confirmation with longitudinal data, including investigation of how hippocampal subfield volumes associate with changes on neuropsychological testing performance over time, will be required [78].

In summary, we present evidence for differential associations between hippocampal subfield volumes and β-amyloid deposition and, increasing age at time of scan and highlight the potential for hippocampal subfield volumes to provide insights into ageing and preclinical AD. Future work focusing on hippocampal subfield morphometry, particularly utilising higher resolution imaging is an important area of future research.

## Acknowledgments

We are very grateful to those study members who helped in the design of the study through focus groups, and to the participants both for their contributions to Insight 46 and for their commitments to research over the last seven decades. We are grateful to the radiographers and nuclear medicine physicians at the UCL Institute of Nuclear Medicine, and to the staff at the Leonard Wolfson Experimental Neurology Centre at UCL. This study is principally funded by grants from Alzheimer's Research UK (ARUK-PG2014-1946, ARUK-PG2017-1946), the Medical Research Council Dementias Platform UK (CSUB19166), and the Wolfson Foundation (PR/ylr/18575). The genetic analyses are funded by the Brain Research Trust (UCC14191). Florbetapir amyloid tracer is kindly provided by Avid Radiopharmaceuticals (a wholly owned

subsidiary of Eli Lilly) who had no part in the design of the study: we are particularly indebted to the support of the late Dr Chris Clark of Avid Radiopharmaceuticals who championed this study from its outset. The NSHD, MR, RH and AW are funded by the Medical Research Council (MC_UU_12019/1, MC_UU_12019/2, MC_UU_12019/3). Some researchers are supported by the NIHR Queen Square Dementia BRU (JMS, NCF), UCL Hospitals Biomedical Research Centre (JMS), Leonard Wolfson Experimental Neurology Centre (JMS, NCF). TDP is supported by a Wellcome Trust Clinical Research Fellowship (200109/Z/15/Z). CHS is supported by an Alzheimer's Society Junior Fellowship (AS-JF-17-011). SJC is supported by an Alzheimer's Research UK Senior Research Fellowship. NCF acknowledges support from the MRC, the UK Dementia Research Institute at UCL, and an NIHR Senior Investigator award, and additional funding from the EPSRC. JMS acknowledges the EPSRC (EP/J020990/1), BHF (PG/17/90/33415), Weston Brain Institute (UB170045), and European Union's Horizon 2020 research and innovation programme (Grant 666992).

## Author Contributions

**Conceptualization:** Thomas D. Parker, Marcus Richards, Nick C. Fox, Jonathan M. Schott.

**Data curation:** Christopher A. S. Lane, Kirsty Lu, Ian B. Malone, Sarah M. Buchanan, Sarah E. Keuss.

**Formal analysis:** Thomas D. Parker, David M. Cash, Jennifer M. Nicholas.

**Funding acquisition:** Thomas D. Parker.

**Investigation:** Christopher A. S. Lane, Kirsty Lu, Sarah-Naomi James, Ashvini Keshavan.

**Methodology:** Thomas D. Parker, David M. Cash, Christopher A. S. Lane, Kirsty Lu, Ian B. Malone, Jennifer M. Nicholas, Sarah-Naomi James, Andrew Wong, Sarah E. Keuss, Carole H. Sudre, Marc Modat, David L. Thomas, Sebastian J. Crutch.

**Project administration:** Heidi Murray-Smith, Andrew Wong, Sarah M. Buchanan, Sarah E. Keuss.

**Software:** Marc Modat.

**Writing – original draft:** Thomas D. Parker.

**Writing – review & editing:** Thomas D. Parker, David M. Cash, Christopher A. S. Lane, Kirsty Lu, Ian B. Malone, Jennifer M. Nicholas, Sarah-Naomi James, Ashvini Keshavan, Heidi Murray-Smith, Andrew Wong, Sarah M. Buchanan, Sarah E. Keuss, Carole H. Sudre, David L. Thomas, Sebastian J. Crutch, Marcus Richards, Nick C. Fox, Jonathan M. Schott.

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
