## [Decision Letter · Decision Letter 0]

22 Aug 2019

PONE-D-19-21691

Hippocampal subfield volumes and pre-clinical Alzheimer’s disease in 408 cognitively normal adults born in 1946

PLOS ONE

Dear Dr Schott,

Thank you for submitting your manuscript to Plos One. Your paper has been reviewed by experts in the field. The reviews have been considered by the Handling Editor and the editorial team. Based on the comments of these reviewers, we feel a major revision would be appropriate. 

Therefore, we invite you to submit a revised version of the manuscript that addresses the points raised during the review process. Please make sure to address ALL comments from the reviewers in a point-to-point manner.

We would appreciate receiving your revised manuscript by Oct 06 2019 11:59PM. To enhance the reproducibility of your results, we recommend that if applicable you deposit your laboratory protocols in protocols.io, where a protocol can be assigned its own identifier (DOI) such that it can be cited independently in the future. For instructions see: http://journals.plos.org/plosone/s/submission-guidelines#loc-laboratory-protocols

We look forward to receiving your revised manuscript.

Kind regards,

Yuankai Huo, Ph.D.

Academic Editor

PLOS ONE

**Journal Requirements**

"NCF’s research group has received payment for consultancy or for conducting studies from Biogen, Eli Lilly Research Laboratories, GE Healthcare, and Roche. NCF receives no personal compensation for the activities mentioned above. JMS has received research funding from Avid Radiopharmaceuticals (a wholly owned subsidiary of Eli Lilly), has consulted for Roche Pharmaceuticals, Biogen, Merck and Eli Lilly, given educational lectures sponsored by GE Healthcare, Eli Lilly and Biogen, and serves on a Data Safety Monitoring Committee for Axon Neuroscience SE."

**Comments to the Author**

1. Is the manuscript technically sound, and do the data support the conclusions?

Reviewer #1: Yes

Reviewer #2: Yes

2. Has the statistical analysis been performed appropriately and rigorously? 

Reviewer #1: Yes

Reviewer #2: I Don't Know

3. Have the authors made all data underlying the findings in their manuscript fully available?

Reviewer #1: Yes

Reviewer #2: Yes

4. Is the manuscript presented in an intelligible fashion and written in standard English?

Reviewer #1: Yes

Reviewer #2: Yes

5. Review Comments to the Author

Reviewer #1: This is a very useful study that evaluated hippocampal subfields among putatively cognitively normal individuals with and without high and low amyloid low. The importance of this. Study lies in the delineation of hippocampal subfields that may be important in understanding early AD pathophysiology. Compared to β-amyloid negative participants (n=334), β-amyloid positive participants (n=74) had lower volume of the presubiculum (3.4% smaller, p=0.012).

This finding with the presubiculum and the fact that this region was not subjected to age effects. like other hippocampal subfields is intriguing.

Minor comments: A better description of specific white matter or other regions for SUVR normalization would be helpful.

Ruling out a formal diagnosis of MCI by clinical evaluation and psychometric tests allows many persons into the study who presumably had neuropsychological impairment without subjective complaints and visa-versa. As such, it not clear whether these participants were actually “cognitively normal” and a number of studies have shown that persons diagnosed with PreMCI, not MC/I not cognitively normal (NACC) or even subjective cognitive complaints may have biological underpinnings. This should. be acknowledged and the authors may even want to run some correlation between the subiculum and other diagnostic tests. The face-name association test might be one such candidate since it was not used in diagnostic formulation.

While I am personally convinced that these are genuine and important findings, multiple tests of statistical significance were run without any reference for the potential of family-wise alpha error or false discovery rate.

Overall, I really liked this paper and thought that it was well argued and well written.

Reviewer #2: The study provided evidence of differential associations in cognitively normal older adults between hippocampal subfield volumes and β-amyloid deposition and, increasing age at time of scan. The relatively selective effect of lower presubiculum volume in the β-amyloid positive group potentially suggest that the presubiculum may be an area of early and relatively specific volume loss in the pathophysiological continuum of Alzheimer’s disease.

Although I do not have any practical knowledge of the performed PET imaging analysis, I believe that the authors did not provide a well-structured, detailed and comprehensive description of the conducted analyses.

Also in the introduction and the discussion, the authors did not provided a comprensive overview of the most relevant literature to the presented work.

Also, it is not clear to me QC of the T1 scans and segmentation

I might have missed this, but were these any figures for results?

6. PLOS authors have the option to publish the peer review history of their article (what does this mean?). If published, this will include your full peer review and any attached files.

Reviewer #1: No

Reviewer #2: No

---

## [Author Response · Author response to Decision Letter 0]

13 Sep 2019

See response to reviewer's file (uploaded)

---

## [Decision Letter · Decision Letter 1]

4 Oct 2019

Hippocampal subfield volumes and pre-clinical Alzheimer’s disease in 408 cognitively normal adults born in 1946

PONE-D-19-21691R1

Dear Dr. Schott,

We are pleased to inform you that your manuscript has been judged scientifically suitable for publication and will be formally accepted for publication once it complies with all outstanding technical requirements.

With kind regards,

Yuankai Huo, Ph.D.

Academic Editor

PLOS ONE

Additional Editor Comments (optional):

All concerns have been addressed

Reviewers' comments:

Reviewer's Responses to Questions

**Comments to the Author**

1. If the authors have adequately addressed your comments raised in a previous round of review and you feel that this manuscript is now acceptable for publication, you may indicate that here to bypass the “Comments to the Author” section, enter your conflict of interest statement in the “Confidential to Editor” section, and submit your "Accept" recommendation.

Reviewer #2: All comments have been addressed

2. Is the manuscript technically sound, and do the data support the conclusions?

Reviewer #2: Yes

3. Has the statistical analysis been performed appropriately and rigorously? 

Reviewer #2: I Don't Know

4. Have the authors made all data underlying the findings in their manuscript fully available?

Reviewer #2: Yes

5. Is the manuscript presented in an intelligible fashion and written in standard English?

Reviewer #2: Yes

6. Review Comments to the Author

Reviewer #2: (No Response)

7. PLOS authors have the option to publish the peer review history of their article (what does this mean?). If published, this will include your full peer review and any attached files.

Reviewer #2: Yes: Guihu Zhao

---

## [Editor Report · Acceptance letter]

8 Oct 2019

PONE-D-19-21691R1 

Hippocampal subfield volumes and pre-clinical Alzheimer’s disease in 408 cognitively normal adults born in 1946 

Dear Dr. Schott:

I am pleased to inform you that your manuscript has been deemed suitable for publication in PLOS ONE. Congratulations! Your manuscript is now with our production department. 

With kind regards,

on behalf of

Dr. Yuankai Huo 

Academic Editor

PLOS ONE